# FourTune: Towards Fully 4-Bit Efficient Post-Training for Diffusion Models

Bowen Xue [* 4]   Zihan Min [* 2]   Xingyang Li [* 2]   Zhekai Zhang [1]   Haocheng Xi [5]   Lvmin Zhang [4]
Maneesh Agrawala [4]   Jun-Yan Zhu [3]   Song Han [2]   Yujun Lin [1]   Muyang Li [1]

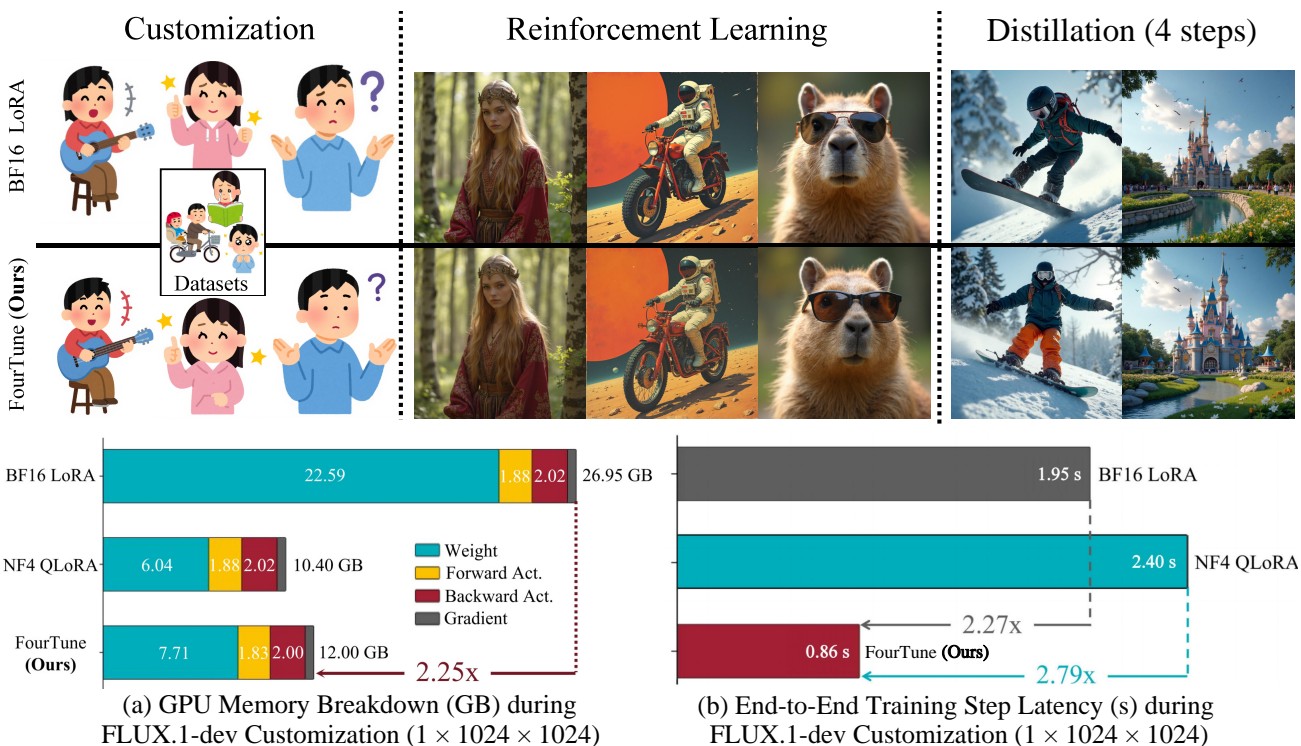

Figure 1. **Qualitative and quantitative comparison of FourTune against baselines. Top:** Visual comparisons across three diverse post-training tasks: Customization, Reinforcement Learning, and Distillation. Despite extremely low-bit quantization (W4A4G4), FourTune produces high-fidelity images visually indistinguishable from the full-precision BF16 LoRA baseline. **Bottom:** Efficiency benchmarks performed on FLUX.1-dev ($1024 \times 1024$). **(a)** FourTune reduces GPU memory consumption by **2.25**$\times$ compared to BF16 LoRA, achieving a compact footprint comparable to NF4 QLoRA. **(b)** In terms of training speed, FourTune significantly outperforms existing methods, achieving **2.27**$\times$ and **2.79**$\times$ speedups over BF16 LoRA and NF4 QLoRA, respectively, effectively breaking the memory-speed trade-off in large model post-training.

## Abstract

Diffusion models have become a dominant paradigm for high-quality generative modeling, while post-training is essential for adapting them to diverse downstream applications. However, post-training of large diffusion models is still challenging due to the prohibitive memory footprints and slow training speed, which existing

parameter-efficient fine-tuning methods only partially address. To overcome these limitations, we propose **FourTune**, an efficient post-training framework for diffusion models based on an end-to-end **W4A4G4** paradigm. FourTune introduces a triple-branch hybrid pipeline that augments the standard LoRA architecture with a frozen numerical stabilizer to isolate quantization-sensitive outliers, enabling stable training under native 4-bit computation. In addition, FourTune employs hardware-efficient block-wise quantization and customized fused kernels to support efficient quantized backpropagation and reduce memory bandwidth overhead. Across customization, rein-

*Equal contribution   [1]Nunchux AI [2]MIT [3]CMU [4]Stanford University [5]UC Berkeley.   Correspondence to: Muyang Li <muyangli@nunchux.ai>.

*Proceedings of the $43^{rd}$ International Conference on Machine Learning*, Seoul, South Korea. PMLR 306, 2026. Copyright 2026 by the author(s).

forcement learning, and distillation tasks, Four-Tune matches the quality of full-precision fine-tuning. On FLUX.1-dev (12B), FourTune reduces memory overhead by **2.25×** and increases end-to-end training throughput by **2.27×** compared to BF16 LoRA.

# 1. Introduction

Generative models have demonstrated remarkable capabilities in synthesizing high-fidelity and semantically complex content (Black Forest Labs, 2024; Qwen Team, 2025). Driven by the pursuit of higher generation quality, the community has scaled up model sizes to unlock greater potential, witnessing explosive parameter growth. Model sizes have increased from the 860M-parameter SD1.5 (Rombach et al., 2022), to the 12B DiT-based FLUX.1 (Black Forest Labs, 2024), and more recently to the 20B-parameter Qwen-Image (Qwen Team, 2025). However, this rapid scaling has substantially raised the computational requirements for both training and deployment, making it increasingly difficult to run such models on consumer-grade GPUs. Although advanced quantization methods (Li et al., 2025; 2023; Shang et al., 2023) have lowered the barrier for inference via low-bit quantization, the resource bottleneck in post-training remains largely unaddressed.

Crucially, post-training serves as a vital pathway toward model practicality and personalization. Specifically, through customization, models can learn specific characters, styles, or concepts (Kumari et al., 2023; Ruiz et al., 2023); via Reinforcement Learning, models can be aligned with human aesthetic preferences (Fei et al., 2025; Shen et al., 2025); and through distillation, the inference steps of diffusion models can be significantly reduced to save deployment costs (Chen et al., 2025; Frans et al., 2025). However, the computation and memory requirements of the diffusion model post-training are still prohibitively high. Consequently, achieving computationally efficient and memory-friendly post-training has emerged as a critical issue.

Recent parameter-efficient fine-tuning (PEFT) methods have gained increasing popularity by substantially reducing the number of trainable parameters and computational cost, while achieving performance comparable to full-parameter fine-tuning (Schulman & Thinking Machines Lab, 2025). However, despite these advances, existing PEFT approaches have not yet translated into further practical efficiency gains, as they remain fundamentally constrained by a persistent memory–speed trade-off. For instance, LoRA (Hu et al., 2022) (Figure 2(a)) reduces training overhead by minimizing the number of trainable parameters. Nevertheless, the full-precision backbone weights still occupy a substantial amount of GPU memory. To alleviate this bottleneck,

QLoRA (Dettmers et al., 2023) (Figure 2(b)) reduces memory usage by quantizing the base model weights. However, it does not reduce computational cost, and the frequent on-line dequantization can even incur additional computational overhead. In summary, a natural question arises: *Is it possible to further accelerate post-training beyond LoRA?*

To address this challenge, we propose **FourTune**, a fully 4-bit post-training framework designed to break existing efficiency bottlenecks. Unlike methods like QLoRA that only quantize weights, FourTune adopts a fully quantized backbone, using 4-bit precision for weights, activations, and gradients (**W4A4G4**) (Figure 2(c)). To the best of our knowledge, FourTune is the **first** fully 4-bit post-training framework for weight, activation, and gradient for large generative models. We introduce a Numerical Stabilizer on top of the standard LoRA fine-tuning framework to ensure stable and convergent training under extremely low-bit quantization regimes. We also employ block-wise quantization to enable efficient on-the-fly transposition for direct 4-bit back-propagation, eliminating transposition and dequantization overhead. Moreover, we propose customized kernel fusion on the LoRA module and the MLP module to minimize memory access overhead. This synergistic design reduces both computational load and memory footprint, achieving memory usage comparable to QLoRA while surpassing the training speed of full-precision LoRA fine-tuning.

We conduct extensive experiments across customization, reinforcement learning, and distillation tasks, showing that **FourTune** enables efficient post-training of diffusion models while matching the generation quality of full-precision fine-tuning. Compared to BF16 LoRA training, FourTune reduces memory footprint by up to 2.25× and accelerates training by up to 2.27×. Overall, FourTune bridges the gap between high efficiency and high performance for diffusion-model post-training.

# 2. Related Work

## 2.1. Post-Training of Diffusion Models

As diffusion models scale up (Podell et al., 2024; Black Forest Labs, 2024; Qwen Team, 2025; Wan Team, 2025), post-training has become essential for downstream adaptation of large diffusion models. This phase typically includes customization, reinforcement learning, and distillation.

Customization enables models to learn new concepts from specific identities, styles, or subjects, while preserving the original generative fidelity (Kumari et al., 2023; Ruiz et al., 2023; Gal et al., 2023). Reinforcement Learning incorporates human preferences into training through online RL (Clark et al., 2024; Prabhudesai et al., 2023; 2024), which performs direct gradient updates on differentiable rewards, and policy-based approaches (Black et al., 2024;

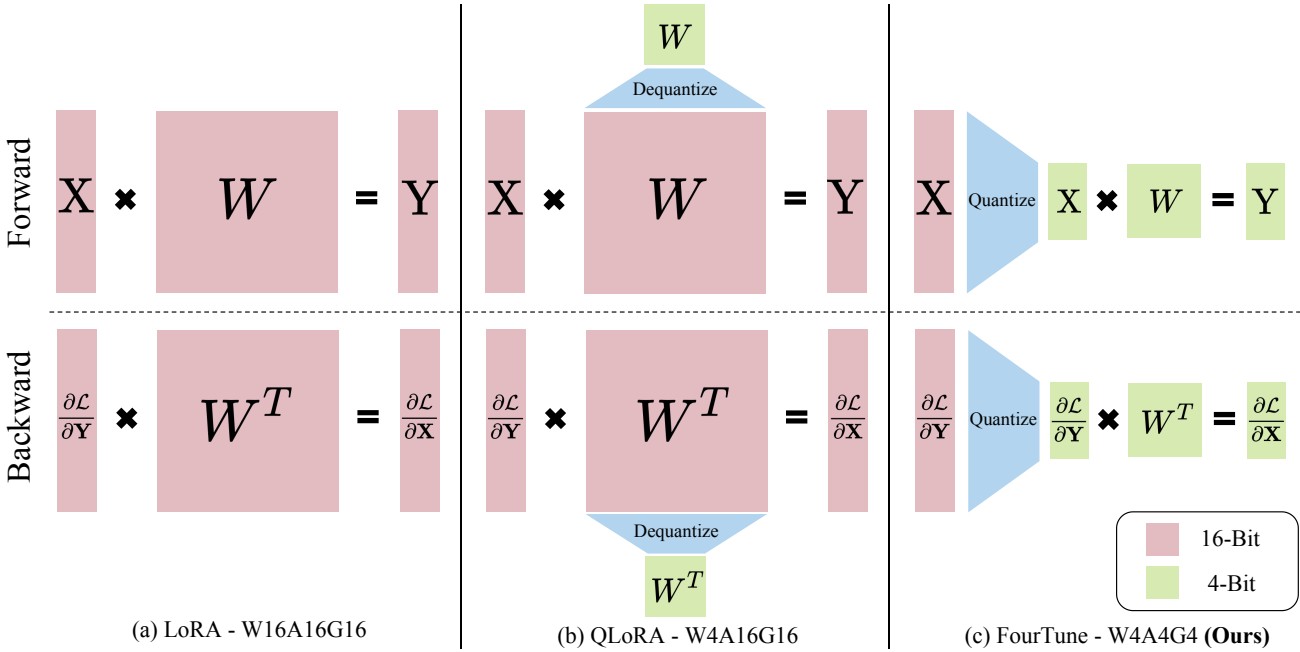

*Figure 2.* Comparison of forward and backward pipelines across different PEFT methods. **(a)** LoRA follows the standard high-precision training pipeline, where weights ($W$), activations ($A$), and gradients ($G$) are all stored and computed in high precision. **(b)** QLoRA reduces the memory footprint by storing weights in 4-bit precision, but still requires on-the-fly dequantization to 16-bit for computation (W4A16G16). **(c)** Our FourTune quantizes weights, activations, and gradients all to 4-bit (W4A4G4), enabling low-bit computation for both the forward and backward passes while preserving memory efficiency.

Fan & Lee, 2023; Fan et al., 2023). Distillation compresses multi-step denoising into efficient few-step generation, using methods like trajectory regression (Luhman & Luhman, 2021; Salimans et al., 2024; Liu et al., 2023), consistency modeling (Song et al., 2023; Gu et al., 2024; Kim et al., 2024), and distribution matching (Yin et al., 2024b;a; Sauer et al., 2024).

### 2.2. Parameter-Efficient Adaptation

Parameter-efficient fine-tuning (PEFT) has become a standard alternative to full fine-tuning, motivated by the observation that adaptation in large models often lies in a low intrinsic dimension (Aghajanyan et al., 2021). Among PEFT methods, Low-Rank Adaptation (LoRA) and its variants (Hu et al., 2022; Soboleva et al., 2025; Zhang et al., 2023; Liu et al., 2024) are particularly effective in both language models and diffusion models (Schulman et al., 2025; YEH et al., 2024; Chen et al., 2025), as they introduce low-rank update matrices while keeping pretrained weights frozen, thereby avoiding additional inference latency incurred by adapter-based methods (Houlsby et al., 2019).

To further improve fine-tuning efficiency, QLoRA (Dettmers et al., 2023) performs LoRA training with a 4-bit quantized backbone, substantially reducing memory consumption at the cost of increased computational overhead due to online dequantization. Building on this, subsequent works ex-

plore improved accuracy–efficiency trade-offs. For instance, LoftQ (Li et al., 2024) tackles this through LoRA-aware quantization techniques, while QA-LoRA (Xu et al., 2024) reduces auxiliary overhead by integrating additional weight to the quantized base model.

### 2.3. Model Quantization

Quantization reduces memory and computation by mapping full-precision weights $W$ to low-bit representations, typically via $\tilde{W} = \text{Round}(W/s)$ with scaling factor $s$. Modern formats such as NF4 and hardware-aware floating-point schemes (e.g., MXFP4 and NVFP4) adopt group-wise scaling to better preserve numerical fidelity on contemporary accelerators (Dettmers et al., 2023; Rouhani et al., 2023; NVIDIA, 2024).

Prior work has shown that diffusion models can be effectively compressed to 8-bit or 4-bit precision for efficient inference (Li et al., 2023; Shang et al., 2023; Li et al., 2025). More recent studies extend quantization to training, including pretraining large language models directly in 4-bit precision (NVFP4) (NVIDIA, 2025). Quantization has also been combined with parameter-efficient adaptation. For example, QeRL (Huang et al., 2025) integrates NVFP4 with LoRA and demonstrates improved exploration in reinforcement learning settings.

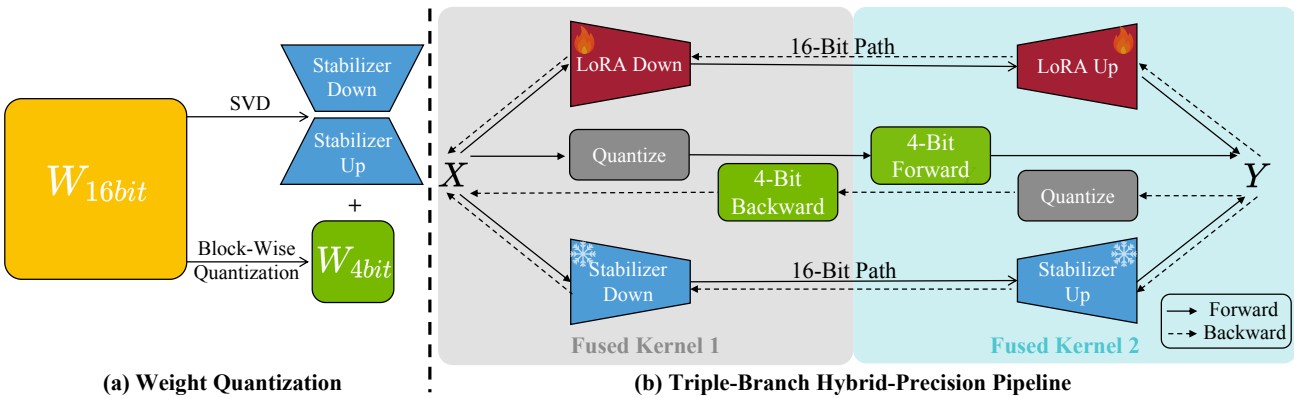

*Figure 3.* **Overview of the FourTune framework.** (a) **Weight Quantization:** High-precision weights are decomposed via SVD into a quantization-friendly 4-bit residual and a high-precision stabilizer. (b) **Triple-Branch Pipeline:** The architecture consists of a frozen 4-bit backbone for arithmetic efficiency, a frozen stabilizer to ensure numerical precision, and a trainable LoRA branch for task adaptation.

## 3. Method

We propose FourTune, a post-training framework that establishes a native 4-bit pipeline for large generative models (Figure 3). Our method quantizes weights, activations, and gradients to 4-bit and executes backbone matrix multiplications directly in 4-bit. This design maximizes arithmetic efficiency while significantly reducing memory footprint. In this section, we describe the overall 4-bit training pipeline backed by a small full-precision stabilizer branch, a block-wise quantization strategy that enables efficient transposition during backpropagation, and customized kernel fusions that improve memory-bandwidth utilization.

### 3.1. Triple-Branch Hybrid-Precision Pipeline

Directly performing forward and backpropagation on aggressively quantized (e.g., 4-bit) weights is often numerically unstable, due to the limited dynamic range and the presence of high-magnitude outliers in weights, activations, and gradients. To address this challenge, we propose a *triple-branch hybrid-precision pipeline* that explicitly decouples numerical stabilization from task adaptation.

Our method features an additional *low-rank, full-precision stabilizer branch* to explicitly handle quantization-sensitive outliers and improve numerical stability during training. Concretely, after isolating the stabilizer component, the remaining frozen backbone weights are quantized using a hardware-friendly low-bit floating-point format (NVFP4) to leverage the powerful 4-bit computing units on GPUs.

Formally, building upon the spectral decomposition from SVDQuant (Li et al., 2025), we decompose the pre-trained weight matrix $W \in \mathbb{R}^{m \times n}$ into a quantization-friendly residual $R$ and a low-rank outlier component $L_{\text{stab}}$:

$$W \approx R + L_{\text{stab}}, \quad \text{where } L_{\text{stab}} = L_1 L_2. \quad (1)$$

Here, $L_{\text{stab}}$ captures the high-magnitude outliers that are

sensitive to quantization, while the residual exhibits a compressed value range suitable for 4-bit representation.

Distinct from previous approaches that utilize this decomposition solely for inference acceleration, we propose to utilize $L_{\text{stab}}$ as a frozen "numerical stabilizer" during training. This allows us to maintain the majority of parameters (the residual $R$) in the 4-bit domain for computationally efficient matrix multiplication (GEMM$_{\text{4bit}}$), while isolating numerical risks in the high-precision branch. To adapt the model to downstream tasks, we introduce a third, trainable low-rank branch $L_{lora}$. The forward propagation for an input $X$ is thus formulated as a triple-branch computation:

$$
\begin{aligned}
Y = \text{GEMM}_{\text{4bit}}(Q(X), Q(R)) \quad &\text{①}\,\text{Frozen 4-Bit Backbone} \\
+ X \cdot L_{\text{stab}} \quad &\text{②}\,\text{Frozen Stabilizer} \\
+ X \cdot (AB) \quad &\text{③}\,\text{Trainable Adapter}
\end{aligned}
$$
$$(2)$$

where $Q(\cdot)$ denotes the 4-bit quantization, applied offline for the residual weights $R$ and dynamically for activations $X$. In this pipeline, while error signals propagate through all branches to ensure correct gradient flow to preceding layers, weight gradients are computed exclusively for the adapter branch $(A, B)$. Branches ① and ② function as frozen, differentiable pathways that facilitate backward propagation without requiring parameter updates. This design enables native 4-bit computation for the backbone, significantly reducing memory bandwidth and computational latency.

### 3.2. 4-Bit Backward Pass via Block-Quantization

In the backward pass, our pipeline minimizes computational redundancy by distinguishing between gradient propagation (required for all branches) and parameter updates (exclusive to the adapter). Let $G = \frac{\partial \mathcal{L}}{\partial Y}$ denote the incoming gradient.

Propagating gradients to the input requires computing $\frac{\partial \mathcal{L}}{\partial X} = GW^\top$. Leveraging the native 4-bit capability of

our backbone, this operation is formulated as:

$$\frac{\partial \mathcal{L}}{\partial \boldsymbol{X}} = \text{GEMM}_{4\text{bit}}(Q(\boldsymbol{G}), Q(\boldsymbol{R})^\top) \quad \text{Backbone Pathway}$$
$$+ \boldsymbol{G} \cdot \boldsymbol{L}_{\text{stab}}^\top \quad \text{Stabilizer Pathway}$$
$$+ \boldsymbol{G} \cdot (\boldsymbol{AB})^\top \quad \text{Adapter Pathway} \tag{3}$$

The critical challenge lies in the term $Q(\boldsymbol{R})^\top$, which requires performing matrix multiplication with a transposed 4-bit weight matrix. Standard per-output-channel quantization is ill-suited for this operation because the scaling factors, originally aligned with the output dimension, become aligned with the reduction dimension (accumulation axis) after transposition. This misalignment prevents the decoupling of scales from the inner product loop, making direct computation inefficient.

To overcome this, we employ a block-wise quantization strategy. By partitioning weights into small, independent blocks, quantization parameters are encapsulated locally. Crucially, this structure facilitates efficient online transposition: since scales are bound to local data blocks rather than global rows or columns, the computation kernel can fetch a 4-bit block and its corresponding scale simultaneously, regardless of whether the access pattern is transposed or non-transposed. This allows us to compute $Q(\boldsymbol{G})Q(\boldsymbol{R})^\top$ directly without dequantizing and re-quantizing the weights, maintaining the high throughput of 4-bit arithmetic.

Concurrently, we compute weight gradients exclusively for the trainable adapter branch. The gradients are derived using standard backpropagation chain rules:

$$\frac{\partial \mathcal{L}}{\partial \boldsymbol{B}} = (\boldsymbol{X}\boldsymbol{A})^\top \boldsymbol{G}, \quad \frac{\partial \mathcal{L}}{\partial \boldsymbol{A}} = \boldsymbol{X}^\top (\boldsymbol{G}\boldsymbol{B}^\top).$$

### 3.3. Kernel Fusion for Efficiency

**LoRA Fusion.** While the stabilizer branch and the trainable low-rank branch introduce negligible additional computation, executing them as independent pathways can incur significant latency due to increased memory access overhead. In this regime, the system bottleneck shifts from computation to memory bandwidth. To mitigate this, we leverage the fact that all three branches share the same input during the quantization and down-projection phases, as illustrated in Figure 3(b). Accordingly, we fuse these operations into a single kernel. Furthermore, since the outputs of these branches must be accumulated, we fuse the 4-bit backbone computation with the up-projection operations to eliminate redundant memory traffic. This strategy substantially reduces the inference latency overhead typically introduced by the LoRA architecture.

**MLP Fusion.** In MLP blocks, intermediate activation tensors often possess high dimensionality (e.g., $3072 \times 12288$).

Applying activation functions such as GeLU on these tensors becomes memory-bound when data is repeatedly written to and read from global memory. To address this, we fuse the first fully connected (FC1) GEMM with the subsequent FC2 quantization kernel. This approach allows intermediate results to be consumed immediately without materializing large tensors in global memory, thereby optimizing bandwidth utilization and accelerating end-to-end execution.

## 4. Experiments

To evaluate the performance and versatility of **FourTune** across different computational tiers, we conduct all experiments on platforms powered by the NVIDIA Blackwell architecture, including $8\times$ NVIDIA RTX Pro 6000 GPUs and $8\times$ NVIDIA RTX 5090 consumer GPUs. Leveraging Blackwell's 4-bit Tensor Cores, we enable a realistic evaluation of both training and inference efficiency.

### 4.1. Experimental Settings

We evaluate FourTune across three core post-training paradigms: Customization, Reinforcement Learning, and Distillation. Please refer to our appendix for more details.

**Customization Setup.** We evaluate FourTune on a hybrid customization benchmark built from Custom Diffusion (Kumari et al., 2023) and web-collected data, covering three tasks: Human Identity, Artistic Style, and General Subject. Experiments are conducted on FLUX.1-dev (12B) (Black Forest Labs, 2024) and Qwen-Image (20B) (Qwen Team, 2025) using LoRA with rank $r = 64$ and identical hyperparameters. Evaluation uses task-specific feature metrics: AntelopeV2 for Human Identity (InsightFace Contributors, 2022; Deng et al., 2019), CLIP for Artistic Style (Radford et al., 2021), and DINOv3 for General Subject (Siméoni et al., 2025), with PyIQA for image quality (Chen & Mo, 2022) and BLIP for prompt following (Li et al., 2022).

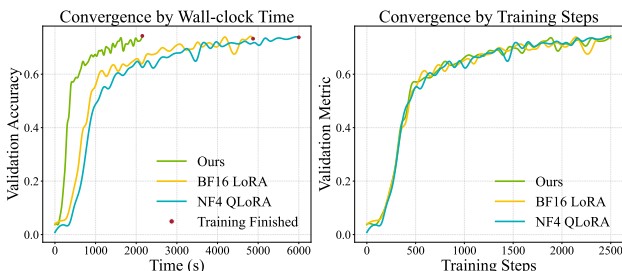

*Figure 4.* Validation curves for customization tasks, showing performance on par with full-precision training and with substantial acceleration.

**Reinforcement Learning Setup.** We evaluate FourTune in a reinforcement learning setting using SRPO (Fei et al.,

*Table 1.* Quantitative results on Customization tasks.

| Task | Method | Precision | FLUX.1-dev | | | | Qwen-Image | | | |
|------|--------|-----------|------------|---|---|---|------------|---|---|---|
| | | | Similarity | Image Quality | Diversity | Prompt Following | Similarity | Image Quality | Diversity | Prompt Following |
| **Identity** | BF16 LoRA | W16A16G16 | 0.771 | 33.49 | **0.577** | **0.941** | 0.601 | 24.54 | 0.742 | **0.998** |
| | FP8 LoRA | W8A8G8 | **0.783** | 30.70 | 0.532 | 0.912 | 0.597 | 21.66 | **0.767** | **0.998** |
| | NF4 QLoRA | W4A16G16 | 0.780 | 32.27 | 0.530 | 0.929 | 0.597 | **25.63** | 0.685 | 0.994 |
| | **Ours** | W4A4G4 | **0.783** | **34.77** | 0.570 | 0.913 | **0.612** | 24.40 | 0.695 | 0.995 |
| **Style** | BF16 LoRA | W16A16G16 | 0.806 | 32.02 | 0.520 | 0.864 | 0.696 | 27.34 | 0.683 | **0.999** |
| | FP8 LoRA | W8A8G8 | **0.821** | 30.05 | **0.605** | **0.893** | 0.692 | 27.29 | 0.676 | **0.999** |
| | NF4 QLoRA | W4A16G16 | 0.806 | 32.19 | 0.529 | 0.883 | **0.710** | **33.72** | 0.646 | 0.928 |
| | **Ours** | W4A4G4 | 0.812 | **34.87** | 0.547 | 0.874 | 0.701 | 29.88 | **0.684** | **0.999** |
| **Subject** | BF16 LoRA | W16A16G16 | 0.731 | **16.57** | 0.415 | **0.946** | 0.618 | 16.92 | 0.707 | **0.999** |
| | FP8 LoRA | W8A8G8 | **0.733** | 15.72 | **0.460** | 0.943 | 0.558 | 18.02 | **0.714** | **0.999** |
| | NF4 QLoRA | W4A16G16 | 0.724 | 15.30 | 0.418 | 0.896 | 0.695 | 15.06 | 0.639 | 0.992 |
| | **Ours** | W4A4G4 | **0.733** | 16.22 | 0.415 | 0.920 | **0.703** | **26.92** | 0.689 | 0.983 |

*Table 2.* Quantitative results for model distillation using the $\pi$-Flow algorithm. We evaluate the 4-step student model on HPSv2 and COCO-10k benchmarks. Performance is measured in terms of teacher alignment (FID), prompt alignment (CLIP), and preference alignment (HPSv2.1). Our W4A4G4 method maintains generation quality comparable to the 16-bit LoRA baseline.

| Method | Precision | HPSv2 prompts | | | COCO-10k prompts | | |
|--------|-----------|---------------|---|---|------------------|---|---|
| | | Teacher align. | Prompt align. | Pref. align. | Teacher align. | Prompt align. | Pref. align. |
| | | FID↓ | CLIP↑ | HPSv2.1↑ | FID↓ | CLIP↑ | HPSv2.1↑ |
| **BF16 LoRA** | W16A16G16 | **15.50** | **0.288** | 0.316 | **5.90** | **0.269** | **0.311** |
| **SparseLoRA** | W16A16G16 | 202.51 | 0.261 | 0.245 | 75.45 | 0.264 | 0.238 |
| **NF4 QLoRA** | W4A16G16 | 15.51 | 0.287 | 0.310 | 6.30 | **0.269** | 0.307 |
| **Ours** | W4A4G4 | **15.50** | 0.283 | **0.317** | 6.70 | 0.266 | 0.310 |

2025), applying preference gradients only to LoRA adapters for parameter-efficient optimization. Experiments follow the original SRPO protocol with FLUX.1-dev as the backbone, trained on HPDv2 and guided by the HPSv2.1 reward model (Wu et al., 2023). Evaluation is performed using Aesthetic Score v2.5 (discus0434, 2024), PickScore (Kirstain et al., 2023), ImageReward (Xu et al., 2023), and SGP-HPS (Fei et al., 2025).

**Distillation Setup.** We evaluate FourTune on model distillation using $\pi$-Flow (Chen et al., 2025), comparing against BF16 LoRA, NF4 QLoRA, and SparseLoRA (Khaki et al., 2025). Following the original $\pi$-Flow protocol, we distill FLUX.1-dev (Black Forest Labs, 2024) into a 4-NFE student in a data-free setting. Evaluation is conducted on COCO-10k and HPSv2 prompts (Wu et al., 2023), reporting Teacher-FID for quality, CLIP (Radford et al., 2021) for prompt alignment, and HPSv2.1 for preference alignment.

## 4.2. Main Results

*Table 3.* Quantitative results on Reinforcement Learning tasks using the SRPO algorithm.

| Method | Aes | PickScore | IR | SGP-HPS |
|--------|-----|-----------|----|---------|
| **Base model** | 6.0135 | 0.2300 | **1.1189** | 0.0015 |
| **BF16 Finetune** | **6.3447** | 0.2316 | 1.0209 | **0.0029** |
| **BF16 LoRA** | 6.1832 | 0.2307 | 1.0735 | 0.0020 |
| **NF4 QLoRA** | 6.2189 | **0.2318** | 1.0931 | 0.0020 |
| **Ours** | 6.3119 | 0.2308 | 1.0152 | 0.0027 |

**Accuracy Results.** For **Customization** (Table 1 and Figure 5), FourTune scales post-training to models with up to 20B parameters. It matches the generation quality of full-precision LoRA across human identity preservation, artistic

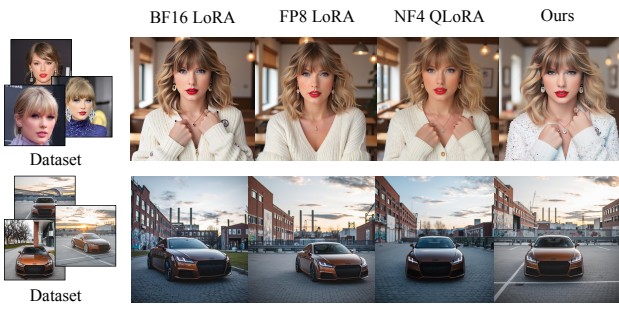

**Customization Comparison**

*Figure 5.* Qualitative comparison on customization. FourTune matches the generation quality of full-precision LoRA.

style transfer, and general subject reconstruction, validating its ability to capture fine-grained visual details.

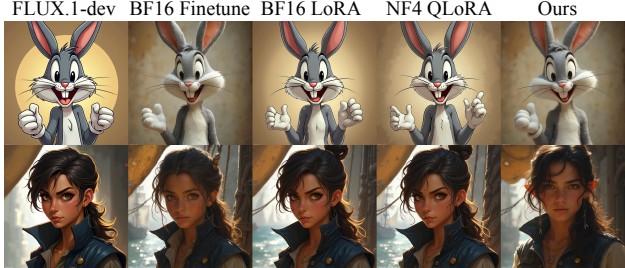

**RL Comparison**

*Figure 6.* Qualitative comparison on RL. FourTune even matches the performance of full-precision, full-parameter finetuning.

For **Reinforcement Learning** (Table 3 and Figure 6), Four-Tune achieves performance comparable to even the full-precision, full-parameter fine-tuning. This indicates that the 4-bit training pipeline retains numerical fidelity to support the fine-grained updates in RL training.

For **Distillation** (Table 2 and Figure 7), FourTune can match the quality of the original 16-bit $\pi$-Flow implementation. Meanwhile, SparseLoRA underperforms FourTune even at 50% sparsity, likely because timestep distillation is highly accuracy-sensitive, whereas SparseLoRA imposes only coarse-grained column sparsity. These results suggest that FourTune remains stable even for distillation, improving efficiency without compromising generation quality.

**Efficiency Results.** We first evaluate the efficiency of FourTune on the DiT component of FLUX.1-dev. By quantizing weights to 4-bit, the weight memory consumption is 7.71 GB, compared to 22.59 GB for BF16 LoRA, achieving a **2.93**× reduction. For a single DiT training step, FourTune achieves a total latency of 612.4 ms whereas BF16 LoRA requires 1541.0 ms, corresponding to a **2.52**× speedup.

We further measure end-to-end training efficiency on the customization task. FourTune reduces the overall memory

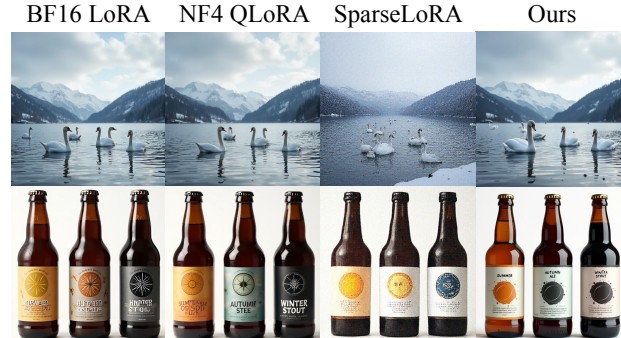

**Distillation Comparison**

*Figure 7.* Qualitative comparison on distillation. FourTune closely matches the visual quality of BF16 LoRA and NF4 QLoRA, while SparseLoRA exhibits noticeable artifacts.

consumption to 14.51 GB, compared to 29.35 GB required by BF16 LoRA, corresponding to a **2.02**× reduction. On NVIDIA RTX 5090, FourTune achieves a single-step training time of 0.86 s, corresponding to a **2.27**× speedup over BF16 LoRA and a **2.79**× speedup over NF4 QLoRA. On NVIDIA RTX Pro 6000, FourTune achieves a single-step training time of 0.66 s, yielding a **2.18**× speedup compared to BF16 LoRA and a **2.85**× speedup over NF4 QLoRA.

These results demonstrate that FourTune not only reduces memory consumption substantially, but also consistently delivers practical end-to-end training acceleration.

### 4.3. Ablation Study

To rigorously evaluate the effectiveness of our proposed method, we conducted ablation studies focusing on four key aspects: training precision configurations, the impact of the Stabilizer, the impact of our block-level quantization, and the efficiency gains from kernel fusion optimizations.

**Impact of Precision Configurations.** We introduce a W4A4G4 training framework that achieves performance comparable to full-precision training. To investigate how the precision of different components affects final training outcomes, we conducted experiments across four different precision settings (W4A4G4, W4A16G4, W4A4G16, and W4A16G16) on RL tasks. Specifically, we performed complete training on a subset of 500 prompts from the HPDv2 Benchmark. It is important to note that among these configurations, only W4A4G4 fully benefits from the acceleration provided by 4-bit computation.

As demonstrated in Table 4, variations in component precision do not lead to performance degradation. Notably, our W4A4G4 configuration enables the model to leverage significant hardware acceleration while maintaining performance alignment with full-precision baselines.

*Table 4.* Ablation study on different precision configurations. Note that W4A4G4 is the only setting that fully utilizes 4-bit acceleration.

| Method | Aes | PickScore | IR | SGP-HPS |
|---|---|---|---|---|
| W4A16G16 | 6.0816 | 0.2352 | **1.2231** | 0.0002 |
| W4A4G16 | 6.0308 | **0.2357** | 1.1982 | 0.0005 |
| W4A16G4 | 6.1386 | 0.2354 | 1.2196 | 0.0001 |
| **W4A4G4 (Ours)** | **6.1779** | 0.2353 | 1.1712 | **0.0008** |

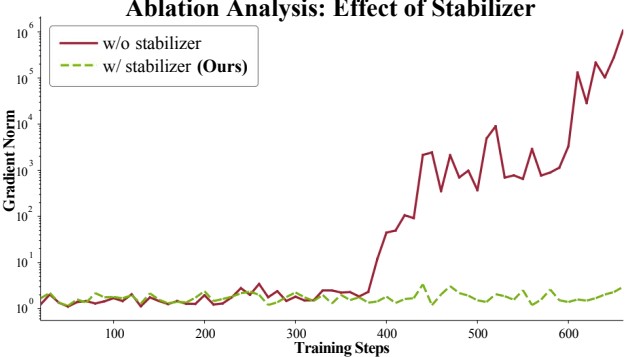

**Ablation Analysis: Effect of Stabilizer**

*Figure 8.* **Effect of the Stabilizer on numerical stability.** Comparison of gradient norms with and without the proposed Stabilizer. The results show that our method prevents the gradient explosion observed in the baseline setting (red line), ensuring stable convergence.

**Effectiveness of the Stabilizer.** During native 4-bit training, the accumulation of quantization errors occasionally triggers gradient explosion, rendering the training process ineffective. To investigate this, we monitored the gradient norms over training steps in knowledge distillation experiments, comparing settings with and without the Stabilizer. As illustrated in Figure 8, the naive W4A4G4 pipeline suffers from gradient explosion, while our FourTune's gradient norms remain stable. These results demonstrate that our Stabilizer effectively maintains numerical stability, thereby preventing gradient explosion and ensuring consistent convergence.

**Impact of Quantization Granularity.** As detailed in Section 3, we adopt block-wise quantization with a $16 \times 16$ granularity to resolve the non-contiguous memory access issues inherent to standard group-wise quantization ($1 \times 16$) during backpropagation. To verify that this coarser granularity does not compromise representation accuracy, we explicitly compare the performance of block-wise versus group-wise strategies on FLUX.1-dev. We measure reconstruction fidelity on 5,000 samples from the MJHQ-30K dataset against the BF16 baseline. As presented in Table 5, our block-wise strategy yields LPIPS and PSNR metrics comparable to the fine-grained group-wise approach. These

results confirm that our design secures significant efficiency gains in the backward pass without incurring any noticeable degradation in generation quality.

*Table 5.* Quantitative comparison of quantization strategies on FLUX.1-dev. We evaluate reconstruction quality on 5,000 samples from the MJHQ-30K dataset compared to the BF16 baseline. Block-wise quantization ($16 \times 16$) maintains high fidelity comparable to the finer-grained group-wise ($1 \times 16$) strategy.

| Quantization Strategy | Granularity | LPIPS ↓ | PSNR ↑ |
|---|---|---|---|
| **group-wise** | $1 \times 16$ | **0.203** | **21.5** |
| **block-wise (Ours)** | $16 \times 16$ | 0.227 | 20.4 |

**Kernel Fusion Breakdown.** To analyze the sources of efficiency gains from Kernel Fusion, we decompose the contributions of individual fusion strategies, as shown in Figure 9. Fusing LoRA-related kernels yields a $1.82\times$ speedup over the unfused 4-bit baseline, while further fusing the MLP components provides an additional $1.10\times$ acceleration. When all kernels are jointly fused, the combined effect results in an overall $2.52\times$ speedup compared to standard LoRA, empirically demonstrating the effectiveness and complementarity of kernel fusion.

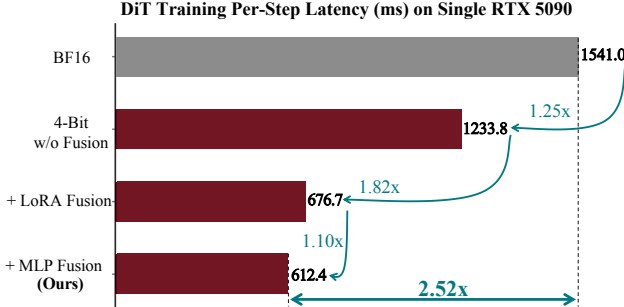

*Figure 9.* Breakdown of efficiency gains from different kernel fusion strategies. The combined optimization results in a $2.52\times$ speedup relative to 16-bit LoRA fine-tuning.

## 5. Conclusion

We have presented FourTune, a fully 4-bit post-training framework that enables end-to-end W4A4G4 optimization for large diffusion models by using a frozen stabilizer branch, block-wise quantization for efficient backward computation, and low-bit fused kernels. Extensive experiments across customization, reinforcement learning, and distillation demonstrate that FourTune matches full-precision baselines while substantially reducing memory footprint and improving throughput. This work contributes to efficient post-training of diffusion models.

## Impact Statement

This work improves the efficiency of post-training large diffusion models through native 4-bit optimization, reducing computational and energy costs and making model adaptation more accessible. However, easier adaptation may also lower the barrier to misuse, such as generating deceptive or harmful content. In our release, we follow the licenses of the base models and datasets, and we encourage downstream users to apply appropriate safety filters and responsible deployment practices.

## Acknowledgements

We thank MIT-IBM Watson AI Lab, MIT and Amazon Science Hub, MIT AI Hardware Program, National Science Foundation, Packard Foundation, Dell, LG, Hyundai Motor Company, and Samsung for supporting this research. We thank NVIDIA for donating the DGX server.

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

# A. Detailed Experimental Settings

## A.1. Customization Evaluation Pipeline

To ensure a comprehensive evaluation, we select five distinct concepts within each category, each exhibiting substantial visual variance. Following standard few-shot customization protocols, we curate five independent prompt–image pairs per concept. This high-variance construction is designed to probe the model's generalization capability and training stability across a diverse set of visual concepts.

During evaluation, we construct five sets of diverse test prompts for each concept. As discussed in Section 4, customization quality is quantified using task-specific feature-space metrics. Specifically, for *Human Identity*, we employ AntelopeV2 to extract facial embeddings and compute cosine similarity with the training samples (InsightFace Contributors, 2022; Deng et al., 2019); for *Artistic Style*, we use CLIP to measure stylistic alignment between generated and reference images (Radford et al., 2021); and for *General Subject*, we adopt DINOv3 to extract fine-grained visual features for evaluating structural and textural consistency (Siméoni et al., 2025). In addition, we assess image quality using PyIQA (Chen & Mo, 2022), measure diversity by averaging pairwise DINOv3 embedding distances among generated images, and evaluate prompt-following behavior using BLIP (Li et al., 2022).

## A.2. Customization Dataset

Figure 10 shows the complete dataset used on customization tasks. The dataset consists of three categories: Human Identity, which focuses on preserving personal identity while allowing variations in appearance or attributes. Artistic Style, which targets adapting artistic or visual styles while keeping the underlying content unchanged. General Subject, which emphasizes generating a specific subject consistently across different contexts.

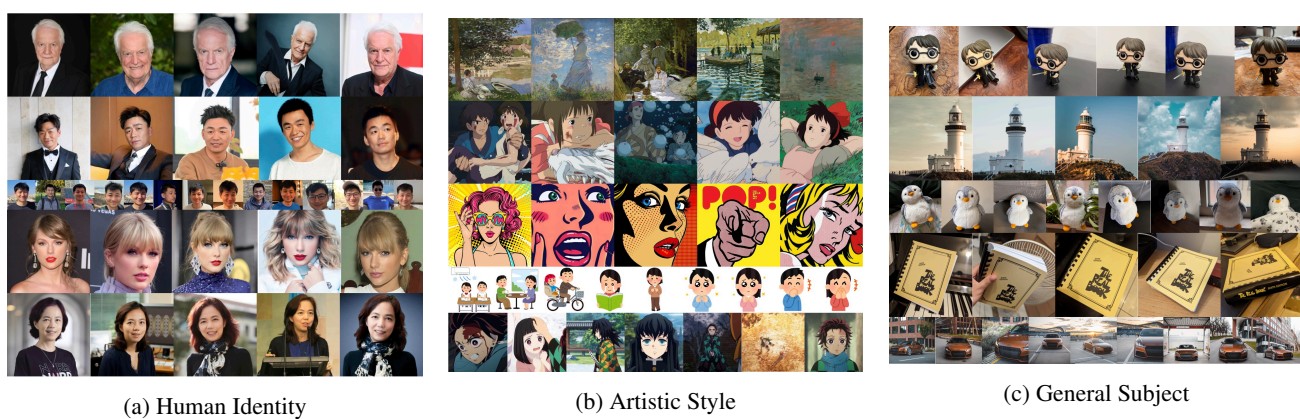

(a) Human Identity        (b) Artistic Style        (c) General Subject

*Figure 10.* Hybrid customization dataset covering three categories: Human Identity, Artistic Style, and General Subject.

## A.3. Customization Metrics

We evaluate customization behavior using two complementary metrics: *diversity* and *Prompt Following*. Diversity is measured using DINO image embeddings to capture visual variation, while Prompt Following is measured using BLIP to assess prompt-image alignment.

**Diversity.** Diversity measures the visual variation among generated images for the same customization task. Given a set of generated images $\mathcal{I} = \{i_1, \ldots, i_n\}$, we extract DINO image embeddings $d(i)$ and compute the average pairwise cosine similarity. The diversity score is defined as

$$D = 1 - \frac{1}{\binom{n}{2}} \sum_{j<k} \cos\big(d(i_j), d(i_k)\big),$$

where larger values indicate higher visual diversity.

**Prompt Following.** Prompt Following evaluates how well generated images follow the input prompts. Given paired prompts and images $\{(p_i, i_i)\}_{i=1}^n$, we compute cosine similarity between BLIP text and image embeddings. The instruction fidelity score is defined as

$$F = \frac{1}{n} \sum_{i=1}^{n} \cos\big(e(p_i), e(i_i)\big),$$

where $e(\cdot)$ denotes BLIP embeddings and larger values indicate better prompt-image alignment.

## B. Generalization

Although our main experiments focus on large-scale DiT backbones, i.e., FLUX.1-dev and Qwen-Image, we further evaluate whether FourTune generalizes beyond the main experimental setup. We consider two complementary settings: transferring to a conventional latent diffusion backbone, SDXL, and running FourTune with an INT4 implementation on an NVIDIA RTX 4090 GPU.

*Table 6.* Generalization to SDXL on identity customization. Our W4A4G4 method maintains generation quality comparable to the 16-bit LoRA baseline.

| Method | Precision | Similarity | Image Quality | Diversity | Prompt Following |
|---|---|---|---|---|---|
| BF16 LoRA | W16A16G16 | **0.454** | 11.17 | 0.658 | 0.935 |
| **Ours** | W4A4G4 | 0.453 | **15.33** | **0.664** | **0.976** |

As shown in Table 6, FourTune transfers well to SDXL under the same W4A4G4 training pipeline. Its overall customization quality remains competitive with the BF16 LoRA baseline, suggesting that the proposed 4-bit training scheme is not restricted to DiT architectures.

*Table 7.* Generalization across quantization formats and hardware/software stacks on FLUX.1-dev identity customization. The INT4 variant is evaluated on an RTX 4090 GPU and achieves a $2.6\times$ speedup over QLoRA.

| Method | Precision | Similarity | Image Quality | Diversity | Prompt Following |
|---|---|---|---|---|---|
| BF16 LoRA | W16A16G16 | 0.771 | 33.49 | **0.577** | 0.941 |
| **Ours** (NVFP4) | W4A4G4 | **0.783** | **34.77** | 0.570 | 0.913 |
| **Ours** (INT4) | W4A4G4 | 0.777 | 31.64 | 0.545 | **0.962** |

FourTune also remains effective under a separate INT4 runtime on the Ada architecture. As shown in Table 7, both NVFP4 and INT4 variants achieve competitive generation quality compared with BF16 LoRA, while the INT4 implementation delivers a $2.6\times$ speedup over QLoRA. These results suggest that FourTune generalizes across model architectures, low-bit formats, and hardware/software stacks.

