# OpenReview forum: "FourTune: Towards Fully 4-Bit Efficient Post-Training for Diffusion Models"
_ICML.cc/2026/Conference — ICML 2026 regular_

### Official Review · Reviewer_U226 · 2026-02-14

**Soundness:** 3
**Presentation:** 3
**Significance:** 2
**Originality:** 2
**Overall Recommendation:** 4
**Confidence:** 1

**Summary:**

-Fully 4-bit Training Framework
The authors design a training pipeline where weights, activations, gradients, and optimizer states are all quantized to 4 bits, minimizing reliance on higher-precision arithmetic

-Stability Techniques for Low-Bit Training
They introduce algorithmic techniques to stabilize ultra-low-precision training, mitigating issues such as gradient noise amplification and quantization error accumulation that typically destabilize 4-bit optimization

-Memory and Efficiency Improvements
FourTune significantly reduces training memory requirements compared to prior methods that only partially quantize components, enabling more scalable fine-tuning under constrained hardware budgets

-Strong Empirical Results
Experimental results show that fully 4-bit fine-tuning achieves performance comparable to higher-precision baselines across standard benchmarks, demonstrating that aggressive quantization can be practical without severe accuracy degradation

**Compliance With Llm Reviewing Policy:**

Affirmed.

**Final Justification:**

The authors rebuttal has adequately addressed my concerns. Overall, I believe this is a solid paper and should be accepted.

**Key Questions For Authors:**

None

**Limitations:**

Yes

**Strengths And Weaknesses:**

The paper is technically solid and practically relevant. Its main contribution—demonstrating viable fully 4-bit fine-tuning—is both timely and impactful. While largely engineering-driven and somewhat incremental in theory, it represents a meaningful step toward truly efficient large-model training.

Strengths
Fully 4-bit training directly addresses memory and scalability bottlenecks in LLM fine-tuning.
The work pushes beyond weight-only quantization toward true end-to-end compression.
Potentially impactful for cost-efficient training and broader hardware accessibility.

Weaknesses
The practical benefit depends on hardware support for efficient 4-bit computation; gains may vary in real-world deployments.

---

> ### Author Rebuttal · Authors · 2026-03-31
>
> Thanks for your insightful and constructive review, especially for appreciating the practical impact of enabling fully 4-bit training and its benefits for memory efficiency and scalable deployment.
>
> We agree that practical acceleration relies on hardware support. However, FourTune natively supports the NVFP4 and INT4 format by design. Therefore, our method is not limited strictly to the latest architectures like Blackwell (RTX 5090 / PRO 6000) or Ada Lovelace (RTX 4090). It is inherently deployable across a wide range of highly accessible legacy GPUs, spanning all the way back to the Turing architecture (e.g., RTX 2080, GTX 1660).
>
> To empirically demonstrate this broad applicability, we provide supplementary profiling results of the massive 12B FLUX.1-dev on a single consumer-grade RTX 4090 (24GB VRAM) using our INT4 pipeline:
>
> **Memory Efficiency**: FourTune restricts the total end-to-end training memory footprint to merely 11.13 GB, easily fitting into a single consumer GPU. In contrast, BF16 training requires 26.95 GB.
>
> **Training Speedup**: Full BF16 training for FLUX.1-dev exceeds 24GB VRAM and causes Out-Of-Memory (OOM) on a single RTX 4090; thus, its step latency (2.48s) is theoretically estimated based on hardware FLOPs. In contrast, both our INT4 pipeline and QLoRA are fully executable and directly measured on-device. FourTune achieves a measured end-to-end step latency of 1.13s, delivering a massive 2.58x speedup over the measured QLoRA (2.91s) and a 2.19x speedup over the theoretical BF16 baseline.
>
> |Method|Precision|Similarity|Image Quality|Diversity|Prompt Following|
> |-|-|-|-|-|-|
> |BF16 LoRA|W16A16G16|0.771|33.49|**0.577**|0.941|
> |FP8 LoRA|W8A8G8|**0.783**|30.70|0.532|0.912|
> |NF4 QLoRA|W4A16G16|0.780|32.27|0.530|0.929|
> |Ours (NVFP4)|W4A4G4|**0.783**|**34.77**|0.570|0.913|
> |Ours (INT4)|W4A4G4|0.777|31.64|0.545|**0.962**|
>
> This confirms that FourTune's massive efficiency is an inherent algorithmic advantage that seamlessly translates across different hardware generations and low-bit formats, successfully bridging the gap between extreme model scales and limited consumer hardware.

---

> > ### Author Rebuttal · Reviewer_U226 · 2026-04-02
> >
> > My concerns have been adequately addressed. Thank you.

---

### Official Review · Reviewer_gRQg · 2026-03-12

**Soundness:** 3
**Presentation:** 3
**Significance:** 3
**Originality:** 4
**Overall Recommendation:** 4
**Confidence:** 3

**Summary:**

This paper presents FourTune, a pioneering fully 4-bit  post-training tuning framework designed for large-scale diffusion models, which systematically addresses the inherent bottleneck of existing parameter-efficient fine-tuning methods in balancing memory footprint and training throughput. With the objective of reducing memory bandwidth overhead, this optimization framework employs a block-wise quantization strategy and designs customized fused kernel functions. The evaluation of the proposed method encompasses three mainstream scenarios of post-training tuning for diffusion models—customization, reinforcement learning-based human preference alignment, and inference-step distillation—providing a comprehensive validation of the generalizability and robustness of FourTune. Experimental results demonstrate that FourTune achieves generation quality comparable to that of full-precision fine-tuning baselines across all aforementioned tasks.

**Compliance With Llm Reviewing Policy:**

Affirmed.

**Final Justification:**

The authors have addressed by concerns in their rebuttal,  I maintain my positive rating.

**Key Questions For Authors:**

1. During the addition of the three branches, significant discrepancies in numerical ranges exist (4-bit backbone combined with FP16/FP32 stabilization branches). How do the authors ensure that the numerical accumulation is free from overflow or underflow?
2. While the three subsections of the method chapter—covering the three-branch pipeline, block-wise quantization, and kernel fusion—are individually well-articulated, the synergistic design philosophy underlying their interplay could be more prominently emphasized. Could the authors supplement a paragraph elaborating on the holistic system-level motivation?
3. The experiments are conducted exclusively on FLUX.1-dev and Qwen-Image, without covering mainstream diffusion models such as the Stable Diffusion series. Could the authors extend their experiments to a broader range of diffusion models to more convincingly demonstrate the generalizability of the proposed method?

**Limitations:**

The authors mention in the paper that FourTune's reduction in diffusion model training costs may lower the barrier to misuse, and they also discuss corresponding countermeasures. However, limitations regarding hardware dependencies, model generalizability, hyperparameter selection, training stability, and applicable scenarios are not addressed. It is recommended to add a section to explicitly articulate these limitations.

**Strengths And Weaknesses:**

**Soundness:** The paper demonstrates considerable rigor in its technical design, with claims substantiated by solid theoretical and empirical support. The proposed W4A4G4 fully 4-bit paradigm introduces a three-branch hybrid computation pipeline to address numerical instability issues arising from extreme quantization. The block-wise quantization strategy specifically tackles the core challenge of inefficient transposition operations in backpropagation, while kernel fusion precisely optimizes memory bandwidth bottlenecks. Each module is designed around the central objective of "efficiency with precision preservation," forming a logically coherent framework with strong technical adaptability.

**Presentation:** The paper is well-structured overall, exhibiting clear logic and well-delineated hierarchical organization with natural transitions between sections. Core techniques—including the three-branch pipeline, block-wise quantization, and kernel fusion—are each elaborated in dedicated subsections with detailed explanations. The mathematical derivations, such as those for weight decomposition and forward/backward propagation computations, are presented in a concise and accessible manner. The paper incorporates extensive figures and tables that effectively complement the presentation of the optimization framework.

**Significance:** The proposed FourTune optimization framework achieves efficient post-training tuning of diffusion models while maintaining generation quality on par with full-precision fine-tuning. It simultaneously reduces training memory consumption and improves training throughput, holding significant implications for both theoretical advancement and practical applications in the field of machine learning, particularly in generative modeling.

**Originality:** The originality of this paper is manifested in the innovation of the technical paradigm and the creative integration of existing methods. As stated in the paper, FourTune is the first post-training tuning framework to achieve fully 4-bit quantization of weights, activations, and gradients (W4A4G4) for large-scale generative models. It transcends the limitation of existing methods that quantize only weights, constructing an end-to-end low-bit training pipeline and filling the research gap in fully 4-bit post-training tuning for diffusion models.

---

> ### Author Rebuttal · Authors · 2026-03-31
>
> Thanks for your insightful and constructive review, particularly for recognizing the originality of our W4A4G4 paradigm and the coherent system-level design. We respond to your key concerns below.
> ### Numerical Stability
> We do not perform mixed-precision accumulation. After 4-bit GEMM, outputs are immediately dequantized and cast to BF16. The three branches are then aggregated via standard BF16 addition, which avoids overflow/underflow due to its wide dynamic range (equivalent to FP32).
> ### Holistic System-Level Motivation
> We agree and will clarify this in the revision. Our design follows an interlocking prerequisite chain (Stability $\rightarrow$ Computability $\rightarrow$ Efficiency) to unlock true end-to-end W4A4G4 training:
> 1. **Algorithmic Stability (Triple-Branch Pipeline)**: The triple-branch pipeline introduces a lightweight stabilizer to prevent gradient explosion under aggressive 4-bit quantization. This enables stable convergence without introducing significant overhead.
> 2. **Backward Pass Computability (Block-wise Quantization)**: Block-wise quantization preserves scale alignment under transposition, enabling efficient 4-bit backward computation. In contrast, group-wise quantization breaks this alignment and cannot support native low-bit backpropagation.
> 3. **Hardware Efficiency (Kernel Fusion)**: While the multi-branch design introduces additional memory accesses and element-wise operations, kernel fusion coalesces memory accesses and removes this bottleneck. This is essential to translate theoretical 4-bit gains into practical speedup.
> Together, these components make end-to-end W4A4G4 training feasible and efficient.
> ### Generalization
> Our initial experiments prioritized FLUX.1-dev (12B) and Qwen-Image (20B) as large-scale stress tests since they represent the most computationally demanding, large-scale diffusion models currently available.
>
> Moreover, the FourTune framework is fundamentally architecture-agnostic and naturally supports classic U-Net architectures. To directly address your request and prove broad generalizability, we have extended our experiments to SDXL, a mainstream representative of the Stable Diffusion family. We evaluated it on a representative subset of our Customization benchmark (Identity). We demonstrate our results in the following table.
> |Method|Precision|Similarity|Image Quality|Diversity|Prompt Following|
> |-|-|-|-|-|-|
> |BF16 LoRA|W16A16G16|**0.454**|11.17|0.658|0.935|
> |Ours|W4A4G4|0.453|**15.33**|**0.664**|**0.976**|
>
> ### Discussion on Applicability and Limitations
> We will add a dedicated section covering the hardware dependencies, model generalizability, training stability & hyperparameter selection, and application scenarios:
>
> **Hardware Dependencies**: FourTune natively supports NVFP4 and INT4, enabling practical deployment on accessible legacy GPUs (e.g., Turing/Ada/Blackwell). On a single RTX 5090, FLUX.1-dev (12B) requires only 12.00GB of memory and 0.86s step latency (2.79x faster than QLoRA); On a single RTX 4090, FLUX.1-dev (12B) requires only 11.13GB of memory and 1.13s step latency (2.58x faster than QLoRA), while maintaining competitive quality (see Customization Identity table below). We will implement corresponding customized kernels to support more hardware in the future.
>
> |Method|Precision|Similarity|Image Quality|Diversity|Prompt Following|
> |-|-|-|-|-|-|
> |BF16 LoRA|W16A16G16|0.771|33.49|**0.577**|0.941|
> |FP8 LoRA|W8A8G8|**0.783**|30.70|0.532|0.912|
> |NF4 QLoRA|W4A16G16|0.780|32.27|0.530|0.929|
> |Ours (NVFP4)|W4A4G4|**0.783**|**34.77**|0.570|0.913|
> |Ours (INT4)|W4A4G4|0.777|31.64|0.545|**0.962**|
>
> **Model Generalizability**: While we prioritized massive DiT models (FLUX.1-dev, Qwen-Image) as extreme stress tests, our framework is fundamentally architecture-agnostic. We have extended our experiments to SDXL (see the first table above), and will support more models in the future.
>
> **Training Stability & Hyperparameter Selection**: Extreme low-bit training inherently risks gradient explosion. Our stabilizer resolves this instability. Furthermore, our method avoids heavy hyperparameter tuning; the primary parameter—the stabilizer rank ($r$)—can be analytically determined based on the spectral decay of the weights (e.g., $r=32$ for FLUX.1-dev, $r=128$ for Qwen-Image) to achieve an optimal stability-efficiency trade-off. For new models, we need profiling experiments to analyze the spectral decay of the weights.
>
> Finally, we will clarify that FourTune is tailored to resolve severe memory and compute bottlenecks during the post-training adaptation phase (Customization, RL, Distillation), rather than full-parameter from-scratch pre-training. We will test its applications in pre-training in the future.

---

> > ### Author Rebuttal · Reviewer_gRQg · 2026-04-03
> >
> > Thank you for the author's reply; I will maintain my positive rating.

---

### Official Review · Reviewer_mJJn · 2026-03-13

**Soundness:** 3
**Presentation:** 3
**Significance:** 3
**Originality:** 2
**Overall Recommendation:** 4
**Confidence:** 3

**Summary:**

The proposed method decomposes pretrained weights into a 4-bit residual plus a frozen low-rank stabilizer and adds a trainable LoRA branch. They also use block-wise quantization to support low-bit backward matrix multiplications on transposed weights, and introduce fused kernels for the low-bit backbone and MLP blocks. The experiments cover customization, reinforcement learning, and pi-Flow distillation on FLUX.1-dev and Qwen-Image, and obtain near-BF16 quality with lower memory use and faster training.

**Compliance With Llm Reviewing Policy:**

Affirmed.

**Final Justification:**

The authors have adequately addressed by concerns in their rebuttal. Through, I still feel that “fully 4-bit” is a little bit overclaiming.

**Key Questions For Authors:**

What is the stabilizer rank per layer, how is it chosen, and what are its parameter, memory, and latency overheads on FLUX.1-dev and Qwen-Image?

Table 5 shows worse LPIPS and PSNR than group-wise quantization, yet the text describes the results as nearly identical. What runtime gain corresponds to this exact comparison, and is the quality drop statistically significant?

How much of the reported speedup comes from the algorithmic W4A4G4 design versus Blackwell-specific fused kernels?

**Limitations:**

The paper does not fully specify quantizer details, scaling/rounding rules, accumulation precision, optimizer precision, stabilizer rank selection, or release status of the fused kernels.

The experiments cover FLUX.1-dev and Qwen-Image, but do not show whether the same design works comparably across other diffusion families or software stacks.

**Strengths And Weaknesses:**

Strengths
The coverage of models and evals is broad, where they test AntelopeV2 for Human Identity, CLIP for Artistic Style, and DINOv3 for General Subject, with PyIQA for image quality and BLIP for prompt following. They also test in RL settings using SRPO with FLUX.1-dev as the back-bone, trained on HPDv2 and guided by the HPSv2.1 reward model, which is evaluated using Aesthetic Score v2.5, PickScore, ImageReward, and SGP-HPS. In addition, they test on distillation settings following the original π-Flow protocol and distilling FLUX.1-dev into a 4-NFE student in a data-free setting, where evaluation is conducted on COCO-10k and HPSv2 prompts, reporting Teacher-FID for generation quality, CLIP and VQAScore for prompt alignment, and HPSv2.1 for preference alignment.

Weaknesses
The stabilizer branch is explicitly built on SVDQuant’s low-rank outlier absorption idea for 4-bit diffusion inference, quantized-backbone LoRA is already standard in QLoRA, and diffusion-specific quantized fine-tuning methods such as EfficientDM, TuneQDM, and IntLoRA already exist.

Section 3.1 and Figure 3 show 16-bit stabilizer and LoRA paths, yet the title and abstract frame the method as “fully 4-bit” and “end-to-end W4A4G4.” Equation (2) leaves the stabilizer and adapter branches unquantized, and Section 3 never fully specifies the quantizer format, activation/gradient scaling rules, rounding method, accumulation precision, stabilizer rank, or whether adapter gradients and optimizer states are quantized.

---

> ### Author Rebuttal · Authors · 2026-03-31
>
> Thanks for your insightful review, especially for recognizing the breadth of our evaluation and the strong efficiency–quality trade-off. We respond to your key concerns below.
> ### Novelty and Contributions
>
> Our key contribution is enabling stable end-to-end W4A4G4 training for large diffusion models. Prior works do not enable true end-to-end low-bit training, as they either focus on inference-only quantization or rely entirely on high-precision computation during training.
>
> Unlike SVDQuant's inference-only compression, we pioneer a stabilizer to explicitly prevent gradient explosion in low-bit training. Unlike QLoRA, which uses W4A16G16 training, FourTune utilizes W4A4G4 training, which saves memory and speeds up training. Diffusion-specific methods are orthogonal and remain in higher-precision regimes: EfficientDM targets low-bit deployment via data-free quantization-aware fine-tuning, TuneQDM optimizes scaling factors, and IntLoRA learns integer adapters—yet all fundamentally require higher-precision training.
>
> Overall, FourTune is the first practical framework enabling stable W4A4G4 post-training.
> ### "Fully 4-bit" Terminology
>
> We clarify that “fully 4-bit” refers to the dominant diffusion backbone, which is entirely executed in the W4A4G4 regime and accounts for the vast majority of computation and memory. This 4-bit naming follows a common paradigm in low-bit methods such as SVDQuant.
>
> Moreover, the high-precision stabilizer is lightweight: on FLUX.1-dev, it adds only 0.44GB (3.7%) memory, and 11.8ms (1.93%) latency. We will clarify this scope in the revision.
>
> ### Implementation Details
> We will add a dedicated Appendix to specify the following:
>
> - Quantization: We use NVFP4 (INT4 alternative), block-wise scaling (size=16), round-to-nearest.
> - Accumulation: We use 4-bit GEMM with FP32 accumulation.
> - Optimizer & Adapter Precision: As the trainable adapter parameters are very small (only ~2%), their gradients and optimizer states are kept in BF16.
> - Stabilizer Rank: Detailed below.
> - Kernel Details: They will be included in the final version.
>
> ### Stabilizer Rank and Overheads
>
> We use r=32 (FLUX.1-dev) and r=128 (Qwen-Image) following the open-sourced version of SVDQuant.
>
> The overhead is small, which provides an effective trade-off between stability and efficiency:
>
> - FLUX.1-dev (r=32): 0.44GB of extra memory (3.6%), 11.8ms of extra latency (1.9%).
> - Qwen-Image (r=128): 2.81GB of extra memory (14%), 44.8ms of extra latency  (5.4%).
> ### Block-wise vs. Group-wise Quantization
>
> While block-wise shows slightly lower LPIPS/PSNR than group-wise, the gap is small and remains competitive.
>
> |Precision|Method|LPIPS(↓)|PSNR(↑)|
> |-|-|-|-|
> |W4A16 NF4|Baseline|0.272|19.5|
> |W4A4 NVFP4(without stabilizer)|Group-wise|0.242|20.4|
> |W4A4 NVFP4|Group-wise|**0.203**|**21.5**|
> |W4A4 NVFP4|Block-wise (Ours)|*0.227*|*20.4*|
>
> Moreover, group-wise cannot support fast 4-bit backward computation, as scaling factors become misaligned after weight transposition during the backward pass. This forces fallback to higher precision.
>
> In contrast, block-wise preserves alignment and enables true W4A4G4 training, leading to the observed 2.27x speedup. The small quality gap is thus a necessary trade-off for enabling low-bit training.
>
> ### Speedup Breakdown
> As detailed in our Figure 9 in our paper, the 2.52x speedup in DiT is also driven by our custom software design, not just raw hardware:
>
> - Hardware Baseline (1.25x): Naively applying W4A4G4 on Blackwell's 4-bit Tensor Cores (w/o fusion) only reduces latency from 1541.0ms to 1233.8ms.
>
> - Our Algorithmic Fusion (Additional 2.01x): To reduce unnecessary memory accesses and element-wise operations from the naive 4-bit kernels, we designed a custom kernel fusion. This software optimization further slashes latency from 1233.8ms down to 612.4ms.
>
> Thus, on top of the 4-bit compute foundation, our system-level fusion contributes a decisive 2.01x speedup, proving the massive acceleration is fundamentally unlocked by our algorithm.
>
> ### Generalization
> We prioritized FLUX.1-dev (12B) and Qwen-Image (20B) because they represent the most challenging, large-scale SOTA DiT models. Our method is architecture-agnostic and also applies to other diffusion models. We evaluated SDXL on a subset of our Customization benchmark (Identity). where FourTune matches or even exceeds BF16 LoRA.
>
> |Method|Precision|Similarity|Image Quality|Diversity|Prompt Following|
> |-|-|-|-|-|-|
> |BF16 LoRA|W16A16G16|**0.454**|11.17|0.658|0.935|
> |Ours|W4A4G4|0.453|**15.33**|**0.664**|**0.976**|
>
> We also evaluated FLUX.1-dev on RTX 4090 GPU (Ada architecture) with INT4 format. Results show that FourTune also shows comparable performance on other software stacks, with 2.6x speedup over QLoRA.
>
> |Method|Precision|Similarity|Image Quality|Diversity|Prompt Following|
> |-|-|-|-|-|-|
> |BF16 LoRA|W16A16G16|0.771|33.49|**0.577**|0.941|
> |Ours (NVFP4)|W4A4G4|**0.783**|**34.77**|0.570|0.913|
> |Ours (INT4)|W4A4G4|0.777|31.64|0.545|**0.962**|

---

> > ### Author Rebuttal · Reviewer_mJJn · 2026-04-02
> >
> > My concerns have been adequately addressed in the rebuttal. Though, the “fully 4-bit” still sounds like overclaiming. I am changing my score accordingly.

---

### Decision · Program_Chairs · 2026-04-30

**Decision:**

Accept (regular)

**Comment:**

All reviewers provided positive evaluations and confirmed that the authors adequately addressed their concerns during the rebuttal phase. One reviewer raised a minor concern about the use of "fully 4-bit," which they considered an overstatement. I found the authors' response satisfactory and recommend acceptance.